# 8a, a New Acridine Antiproliferative and Pro-Apoptotic Agent Targeting HDAC1/DNMT1

**DOI:** 10.3390/ijms22115516

**Published:** 2021-05-24

**Authors:** Qiting Zhang, Ziyan Wang, Xinyuan Chen, Haoxiang Qiu, Yifan Gu, Ning Wang, Tao Wang, Ze Wang, Huabin Ma, Yufen Zhao, Bin Zhang

**Affiliations:** 1Institute of Drug Discovery Technology, Ningbo University, Ningbo 315211, China; 1811075025@nbu.edu.cn (Q.Z.); 1911074031@nbu.edu.cn (T.W.); mahuabin@nbu.edu.cn (H.M.); 2Li Dak Sum Yip Yio Chin Kenneth Li Marine Biopharmaceutical Research Center, College of Food and Pharmaceutical Sciences, Ningbo University, Ningbo 315800, China; wzy18858014311@163.com (Z.W.); chenxinyuan2001@163.com (X.C.); qhx1841094309@163.com (H.Q.); 196001559@nbu.edu.cn (Y.G.); 2011085073@nbu.edu.cn (Z.W.); 3Key Lab of Bioorganic Phosphorus Chemistry & Chemical Biology, Department of Chemistry, Tsinghua University, Beijing 100084, China

**Keywords:** HDAC1, DNMT1, apoptosis, anti-proliferative, epigenetic, acridine

## Abstract

Epigenetic therapy using histone deacetylase (HDAC) inhibitors has become an attractive project in new drug development. However, DNA methylation and histone acetylation are important epigenetic ways to regulate the occurrence and development of leukemia. Given previous studies, *N*-(2-aminophenyl)benzamide acridine (**8a**), as a histone deacetylase 1 (HDAC1) inhibitor, induces apoptosis and shows significant anti-proliferative activity against histiocytic lymphoma U937 cells. HDAC1 plays a role in the nucleus, which we confirmed by finding that **8a** entered the nucleus. Subsequently, we verified that **8a** mainly passes through the endogenous (mitochondrial) pathway to induce cell apoptosis. From the protein interaction data, we found that **8a** also affected the expression of DNA methyltransferase 1 (DNMT1). Therefore, an experiment was performed to assess the binding of **8a** to DNMT1 at the molecular and cellular levels. We found that the binding strength of **8a** to DNMT1 enhanced in a dose-dependent manner. Additionally, **8a** inhibits the expression of *DNMT1* mRNA and its protein. These findings suggested that the anti-proliferative and pro-apoptotic activities of **8a** against leukemia cells were achieved by targeting HDAC1 and DNMT1.

## 1. Introduction

The reversible acetylation of histones is mostly maintained through the physical and functional interplay between histone acetyltransferases (HATs) and histone deacetylases (HDACs), the latter of which were identified as valuable targets for anticancer drug design [1]. HDAC activity inhibition has been proven to be an effective strategy for treating tumors [2]. Treatment of leukemia cells with HDAC inhibitors can cause cell growth arrest, inhibit angiogenesis, induce cell differentiation and apoptosis, and enhance immunity [3,4]. HDAC inhibitors, such as trichostatin A (TSA), valproic acid (VPA), suberoylanilide hydroxamicacid (SAHA), MS-275, and benzamide (chidamide), have become a new class of drugs in cancer therapy [5]. Class I HDACs are most closely related to the development of tumors, including the histone deacetylase 1 (HDAC1) subtype. HDAC1 is widely involved in various protein complexes’ catalytic processes, affecting the proliferation, metastasis, differentiation, infiltration, gene transcription, and other tumor cell processes [6]. Members of the HDAC family of nuclear enzymes can remove acetyl groups from histones to pack chromatin and repress transcription [7]. They can also induce mitochondrial damage and interfere with energy metabolism [8]. The downregulation of HDAC1 can enhance chronic lymphocytic leukemia cells’ sensitivity to tumor necrosis factor ligand-induced apoptosis [9]. It has been reported that HDAC inhibitors can activate either one or both cell death pathways in many cancer models [10]. Given the multitude of cellular effects triggered by HDAC inhibitors, it is probable that several different mechanisms contribute to their anticancer activity [11].

Histone deacetylation represses the transcription of genes responsible for cell differentiation/death. Similarly, aberrant DNA methylation often silences the transcription of tumor-suppressor genes and is considered a hallmark of myeloid neoplasms [12]. The crosstalk between DNA methylation and histone modifications has been reported widely [13]. The combined clinical use of histone deacetylase inhibitors (HDACi) and a DNA methylation agent (HMA) enhances the efficacy [14]. It was found that DNA methyltransferase 1 (DNMT1) mediates DNA methylation to produce changes in chromatin status, and silence tumor suppressor genes, depending on HDAC1 activity [15]. In chronic myelogenous leukemia, a high level of DNA methylation and a low level of histone acetylation lead to drug resistance [16]. Susanna F. Greer found that downregulation of DNMT1 and HDAC1 expression in ovarian cancer cells induced by overexpression of RGS10 could inhibit tumor drug resistance [17]. The links between DNA methylation and histone modification provide opportunities for combination therapies targeting both epigenetic mechanisms [18].

Many preclinical studies have shown that HDAC inhibitors have synergistic effects with other anticancer or cytotoxic agents and can be used in combination with radiation therapy [19]. Such studies provide a basis for the development of dual-targeted inhibitors. Guerrant et al. (2012) have synthesized dual-acting histone deacetylase and topoisomerase II inhibitors. These dual-acting agents are derived from SAHA and anthracycline daunorubicin, respectively. Additionally, these agents potently inhibit the proliferation of representative cancer cell lines [20]. Molecular targeting compounds can specifically interact with their molecular targets, enhance the antitumor effects of treatments, and effectively reduce the toxicity and side effects in the treatment process [21]. For example, 8u, a novel multi-benzylamino acridone derivative, showed low toxicity in mutagenic, tumorigenic, irritant, and reproductive aspects [22].

In our previous work, 16 new *N*-(2-aminophenyl) benzamide acridine analogs (**8a**–**8p**) were synthesized and biologically evaluated as inhibitors of HDAC1 and Topo I [23]. As molecular targeting compounds, they have effective selective inhibition. Among these screening molecules, **8a**, *N*-(2-aminophenyl)-4-(((4-methylacridin-9-yl) amino) methyl) benzamide (Figure 1), which has green fluorescence and an emission wavelength of 476 nm, showed the most significant antiproliferative activity against leukemia cells which was associated with histone 3 acetylation concentration. This study investigated the molecular mechanism of **8a** inhibiting tumor cell growth and analyzed other potential targets. Studies have shown that **8a** as an HDAC1 inhibitor may combine with different pathways to inhibit tumor cells’ growth.

## 2. Results

### 2.1. Antiproliferative Assay and Nuclear Localization

To verify the anti-proliferative activity of **8a**, we used an MTT assay in this study and found that **8a** had a time-dependent effect against U937 cells (Figure 2A) and CCRF-CEM, K562 cells (Appendix A). It is worth noting that 2 μM of **8a** within 48 h showed a significant inhibitory effect, and the fluorescence of **8****a** can be well detected at this dose. Thus, we chose a 2 µM concentration of **8a** and 48 h for the following research. HDACs localize in the nucleus to regulate gene transcription and play a role in neurogenesis, apoptosis, and plasticity [24]. HDAC1 contains a C-terminal nuclear localization signal (NLS) but lacks a nuclear export signal (NES); it exists in the nucleus [25]. Since **8a** has green fluorescence, we used a fluorescence microscope to identify the drug action site of **8a**. Additionally, we stained the nuclei with 4′,6-diamidino-2-phenylindole (DAPI), making them blue; the membranes were stained with 1,1′-dioctadecyl-3,3,3′,3′-tetramethylindocarbocyanine perchlorate (DIL), making them red. The results showed that the green fluorescence of **8a** coincided with the blue fluorescence of nuclear staining, which indicates that **8a** can indeed enter the nucleus (Figure 2B). HDACs serve to remove acetyl groups from histones and thereby repress transcription. Studies have shown that SAHA can augment histone 3 acetylation by inhibiting HDACs. Moreover, it has been shown that valproic acid can prevent the reduction of acetyl-histone H3 (Ac-H3) in the NEP promoter region and reduce the downregulation of NEP by inhibiting the activation of HDAC1 [26,27]. Therefore, we investigated the effect of **8a** on HDAC1 by detecting the expression of Ac-H3 and total H3. A previous study revealed that **8a** could affect Ac-H3 in a concentration-dependent manner in leukemia cells [23]. And our experiment showed the same result of Ac-H3 but no effect on total H3 (Appendix A). Chidamide (CS055) is a novel benzamide class of histone deacetylase inhibitor [28]. We analyzed Ac-H3 expression in cells treated by **8a** and chidamide (the isoform-selective HDAC inhibitor) and found that the combination of **8a** and chidamide caused a high expression of Ac-H3 (Figure 2C,D). The above results illustrated that **8a** can enhance the histone inhibitory activity of chidamide, which further shows that **8a** and chidamide have synergistic effects.

### 2.2. Inducing Apoptosis

Studies have shown that HDAC1 inhibitors can induce cell apoptosis, cause mitochondrial damage, and interfere with energy metabolism [8,9]. We found that **8a** can induce the apoptosis of leukemia cells. In order to detect the pathway of apoptosis induced by **8a**, we performed a series of apoptosis-related experiments on three types of human leukemia cells, including U937, K562, and CCRF-CEM cells. Firstly, we detected apoptosis induced by **8a** using the Annexin V-FITC apoptosis detection kit. The flow cytometry results showed that the number of U937 apoptotic cells increased gradually in a time and dose-dependent manner (Figure 3A). However, the same result was observed in K562 cells and CEM cells, but the effect is weaker than U937 cells (Appendix A). Moreover, the decrease of mitochondrial membrane potential is a landmark event in the early stage of apoptosis. Therefore, we tested whether **8a** caused mitochondrial damage via mitochondrial membrane potential and reactive oxygen species test, successively. 5,5′,6,6′-Tetrachloro-1,1′,3,3′-tetraethylbenzimidazolyl-carbocyan ine iodide (JC-1) is a mitochondrial probe. The transformation of JC-1 from red fluorescence to green fluorescence was used as an indicator of early apoptosis [29]. Carbonyl cyanate-3-chlorophenylhydrazone (CCCP) is a proton carrier that induces proton gradient decoupling in the mitochondrial intima, causes the loss of membrane potential on both sides of the mitochondrial inner membrane, and finally, induces apoptosis. [30]. Here, CCCP was used as the positive control, and JC-1 was used to label the early apoptotic cells to detect the effect of **8a** on the mitochondrial membrane potential. Mitochondrial membrane potential experiments showed that normal cells’ red fluorescence was reduced, and the green fluorescence of apoptotic cells was enhanced, both in a dose-dependence manner. These data indicate that **8a** caused mitochondrial damage by the collapse of the mitochondrial membrane potential (Figure 3B). Similar results were found in CCRF-CEM and K562 cells (Appendix A). The production of reactive oxygen species (ROS) is closely related to the development of tumor cells. It has been reported that a high level of ROS production can induce the apoptosis of tumor cells [31]. Eot-Houllier et al. (2009) demonstrated that ROS levels are elevated after treatments with HDAC inhibitors, and that HDAC inhibitors can activate cell death after ROS-induced DNA damage [32]. The reactive oxygen species test results showed that **8a** was able to enhance mitochondrial ROS of U937 cells (Figure 3C). Similarly, 8a could also enhance the mitochondrial ROS of CCRF-CEM and K562 cells (Appendix A), but the effect on K562 cells was significantly weaker than the other two tumor cells. In summary, as an inhibitor of HDAC1, **8a** can induce apoptosis, and damage mitochondrial function.

Subsequently, we explored the pathway of apoptosis induced by **8a**. Caspase-9 is involved in the activation cascade of caspases and is responsible for the execution of apoptosis. This mitochondria-dependent caspase activation mechanism is referred to as the “intrinsic” pathway of apoptosis [33]. Caspase-8 is involved in death receptor-mediated exogenous programmed cell death and various apoptotic stimuli [34]. Therefore, we added different concentrations of **8a** (0.5, 1,2, 5, and 10 μM) to caspase-8 and caspase-9 knockout cells to detect cell vitality. The experimental results showed that in caspase-9 knockout cells, the inhibitory effect of **8a** on cell activity was weakened, but this phenomenon was not found in caspase-8 knockout cells. Interestingly, in caspase-8 and caspase-9 double knockout cells, the cell viability was enhanced under the influence of **8a**. These results indicate that **8a** mainly affects the endogenous pathway mediated by mitochondria, and its regulation of the exogenous pathway depends on the mitochondrial pathway (Figure 4A). Z-VAD-FMK is a well-known pan caspase inhibitor [35]. Therefore, we used Z-VAD-FMK as a positive control to verify whether **8a** affected cell apoptosis through a mitochondria-dependent caspase activation mechanism. CCK8 results showed that the effect of **8a** on the cells that were inhibited by Z-VAD-FMK was significantly reduced (Figure 4B), which further suggested that **8a** induced apoptosis through an endogenous mitochondria-dependent caspase pathway. One pathway of caspase activation is triggered by the release of cytochrome C from the mitochondria into the cytoplasm. In this mitochondrial death pathway, the expression ratio of pro-apoptotic Bax protein to anti-apoptotic Bcl-2 protein ultimately determines cell death or survival [36,37]. Consequently, we analyzed the basal expression of apoptotic protein markers. It was indicated that **8a** could increase the expression of pro-apoptotic Bax protein and cytochrome C in U937 cells. In the meantime, **8a** significantly decreased the expression of the anti-apoptotic protein Bcl-2 (Figure 4C–F), and the expression was similar in CCRF-CEM cells and K562 cells (Appendix A). In conclusion, it was revealed that **8a** induces apoptosis through endogenous (mitochondrial) pathways.

### 2.3. Proteomics Analysis

The molecular mechanism of **8a** inducing apoptosis was further explored through complete proteomics analysis. Firstly, by setting the threshold for the adjusted *p* value as <0.05 and that for log2 fold change (FC) as >1 for differential expression analysis, we screened out the differential proteins. The BP (biological process), CC (cell component), and MF (molecular function) of GO analysis results were obtained through the Functional Annotation Tool in DAVID Bioinformatics Resources 6.8 (https://david.ncifcrf.gov/summary.jsp (accessed on 25 March 2021)). The results showed that the main sites of action of **8a** are in the nucleus and mitochondria, which are related to apoptosis and the lifecycles of tumor cells (Figure 5A-C). Interestingly, many differential proteins were found to be associated with HDAC1 and DNMT1 at the same time. DNA methylation and histone acetylation are the main methods of epigenetic regulation for tumors [15]. For this reason, we used a Venn diagram (Appendix A), PPI (protein–protein interactions, Figure 5D), and a heatmap (Figure 5E) to analyze these proteins. Proteins associated with HDAC1 and DNMT1 were searched for through the “Single Protein” area by name/identifier on STRING (https://www.string-db.org (accessed on 25 March 2021)). The Venn diagram shows that a total of 19 proteins were associated with both HDAC1 and DNMT1. The interaction networks of these 19 proteins were shown separately through protein–protein interactions (PPI). This also revealed that **8a** was closely related to the pathways regulated by HDAC1 and DNMT1. Further heatmap analysis of the differential proteins showed that **8a** down-regulated not only HDAC1, but also DNMT1. Therefore, we further verified that **8a** targets DNMT1.

### 2.4. **8a**-DNMT1 Molecular Docking and Interactions

DNMT1 inhibitors are mostly nucleoside analogues. However, nucleoside DNMT1 inhibitors have a strong binding effect on DNA, which can easily cause poisoning and side effects. Therefore, the development of DNMT1 inhibitors is gradually turning to non-nucleoside drug modification. The emergence of SGI-1027 (*N*-(4-(2-amino-6-methylpyrimidin-4-ylamino) phenyl)-4-(quinolin-4-ylamino) benzamide) provides a theoretical basis for the development of non-nucleoside DNMT1 inhibitors, but its cell targeting and inhibitory activity are not strong [38,39]. Here, we used computer molecular simulation technology to detect the action modes of **8a** and DNMT1 (PDB ID:4WXX), and SGI-1027 was used as a control. Molecular docking (Figure 6A–D) showed that the tricyclic acridine system, as a hydrophobic region, was more capable of forming multiple Pi bonds with the amino acid residues of DNMT1 protein than the bicyclic quinoline ring. The 9-amino group on the acridine ring effectively participated in the formation of a hydrogen bond with the amino acid residues Asn1578 of the target protein. In contrast, the 4-quinoline amino group in the SGI-1027 structure did not participate in the hydrogen bond. In addition, benzamide replaces pyrimidine with two hydrogen bonds with Gly1223 and Asn1578, whereas SGI-1027′s pyrimidine group is involved in only one hydrogen bond. Therefore, DNMT1 had better binding with **8a** than with SGI-1027.

Binding affinity is one of the most important indexes for evaluating the binding effect. We examined the binding specificity and affinity of **8a** and DNMT1, using biolayer interferometry (BLI). DNMT1 protein with a His marker was loaded on an NTA sensor chip. **8a** was bound to DNMT1 (50 μg/mL) with different concentration gradients. The kinetic curves of association and dissociation are shown in Figure 6E. Analyses of the resulting binding kinetics revealed that **8a** displayed a high binding affinity with DNMT1, and the Kd (dissociation constant) value between **8a** and DNMT1 protein was 1.90 nM.

### 2.5. **8a** Affected the Expression of DNMT1

We also detected that **8a** affected the expression of DNMT1 in three types of cells. The mRNA expression of *DNMT1* was analyzed by quantitative RT-PCR. The result showed that **8a** downregulated the expression of *DNMT1* in a dose-dependent manner (Figure 7A). Simultaneously, we also found that **8a** could affect the protein expression of DNMT1 in U937 cells (Figure 7B,C). It shows the same trend in CCRF-CEM and K562 cells (Appendix A). Further, we combined **8a** with SGI-1027 to observe the effect of the combination by Western blots. The Western blots results showed that compared with monotherapy, the combination enhanced the inhibition of DNMT1. In addition, **8a** inhibited the expression of DNMT1 more strongly than SGI-1027 (Figure 7D,E). In conclusion, our results illustrate that the mRNA and protein levels of DNMT1 will reduce gradually as the **8a** dose increases, and at the same time, the combined use of **8a** and SGI-1027 improved the effect. This suggests that **8a** as an HDAC1 inhibitor can target DNMT1, and can synergistically work with SGI-1027.

## 3. Discussion

Epigenetic silencing is frequently observed in cancer, which leads to the development of leukemia cells. Histone deacetylases regulate non-nuclear chromatin-dependent cells’ abnormal proliferation, leading to drug resistance in leukemia cells [40]. HDAC inhibitors show low toxicity in transformed cells and could synergistically enhance the anticancer activity of multiple chemotherapy drugs [41]. Class I HDACs are strongly correlated with the development of tumors. Research shows that the downregulation of HDAC1 can enhance the sensitivity of chronic lymphocytic leukemia cells to tumor necrosis factor ligand-induced apoptosis [42]. In previous studies, **8a**, as an inhibitor of HDAC1 and Topo I, showed the most significant antiproliferative activity against leukemia cells. Therefore, we tested the inhibition rate of **8a**-treated cells in different concentrations and a gradient of treatment times. Our studies manifested that 8a could increase the expression of Ac-H3 in U937 cells, and this effect was enhanced when combined with chidamide.

HDAC inhibitors have been reported to have several different mechanisms contributing to their anti-tumor activity, such as inducing mitochondrial dysfunction, necroptosis, apoptosis, and interfering with energy metabolism [43]. To verify these results, we explored the pathway of **8a** affecting cell apoptosis. *Caspase*-8 and *caspase*-9 are very important genes in exogenous and endogenous apoptotic pathways, respectively [44]. We used **8a** to act on *caspase*-8 and *caspase*-9 gene knockout cells, and it was found that the inhibitory effect of 8a on cell activity is mainly related to the caspase-9 pathway, and its effect on the caspase-8 pathway depends on the caspase-9 pathway. Simultaneously, we used Z-VAD-FMK (nonretrograde caspase inhibitor) to restrain the caspase pathway, and we demonstrated that **8a** acts mainly via an endogenous mitochondria-dependent caspase pathway to induce apoptosis. Further, we also found that **8a** could increase the expression of cytochrome C, and influence the expression ratio of pro-apoptotic Bax protein to anti-apoptotic Bcl-2. Overall, the above experiments showed that **8a** induced apoptosis mainly through the mitochondrial endogenous pathway. Lastly, we analyzed how **8a** affects the pathways of cell development by proteomics and bioinformatics. GO analysis results displayed that **8a** induced cell death through several pathways, especially the mitochondrial apoptotic pathway. Additionally, from the Venn diagram, PPI and heatmap analyses, we found that **8a** down-regulated not only *HDAC1* genes, but also *DNMT1*. Therefore, we verified whether **8a** targeted DNMT1.

It has been reported that DNMT is overexpressed in a variety of leukemias (especially multi-drug resistant leukemias) and is directly related to the poor prognosis of tumors [37]. Compared with other DNA methyltransferases (such as DNMT3A), DNMT1 subtypes are not susceptible to mutation and are more suitable as targets for anti-tumor drugs [45]. Simultaneous inhibition of DNMT1/HDAC1 can play a synergistic role in improving anti-tumor drugs’ therapeutic effects and reducing the incidence of drug resistance [46]. More importantly, nucleoside DNMT1 inhibitors have a strong binding effect on DNA, which can easily cause poisoning and side effects. SGI-1027, as a non-nucleoside DNMT1 inhibitor, provided a theoretical basis for the development of this class of inhibitors but did not show good inhibitory activity. **8a** is expected to be a good non-nucleoside DNMT1 inhibitor, and has shown more significant inhibitory activity than SGI-1027. We used a computer simulation prediction to find that the effect of **8a**’s binding with DNMT1 was better than that of SGI-1027. The BLI measurement showed that the binding strength between **8a** and DNMT1 increased in a dose-dependent manner. Furthermore, we verified the effect of **8a** on both mRNA and protein-level expression of DNMT1, thereby finding that the mRNA and protein expression levels of DNMT1 decreased with the concentration of **8a**, and the inhibition effect was stronger than that of SGI-1027. At last, we combined **8a** with SGI-1027, and found the combination would heighten the effect. These results proved that **8a**, as an HDAC1 inhibitor, could inhibit the expression of DNMT1. Additionally, it has a synergistic effect with SGI-1027.

Epigenetic silencing of tumor suppressor genes (TSGs) through DNA methylation and histone changes (acetylation) is the main hallmark of cancer [8]. To date, six agents in two epigenetic target classes (DNMT1 and HDAC1) have been approved by the US Food and Drug Administration (FDA) [47]. Current studies suggest that agents that intervene with this process by “turning back on” silenced genes could represent a significant advancement in treating many types of cancer [47,48]. Therefore, we speculate that **8a** causes cancer cells to undergo apoptosis through the re-expression of tumor suppressor genes. However, the specific mechanism of action needs further study.

## 4. Materials and Methods

### 4.1. Cell Culture

The *caspase*-8 and *caspase*-9 knockout HeLa cells were provided by Dr. Huabin Ma (Institute of Drug Discovery Technology, Ningbo University, Ningbo, China). The U937, CCRF-CEM, and K562 cell lines were purchased from the American Type Culture Collection (ATCC, Manassas, VA, USA). Cell lines were cultured in RPMI-1640 and IMDM medium (Hyclone) supplemented with 10% fetal bovine serum (BI), 100 U/mL penicillin, and 100 μg/mL streptomycin.

### 4.2. Viability Assay

The MTT assay determined cell viability. The cell number in the logarithmic phase was 3 × 10^5^–6 × 10^5^/mL. It was treated with **8a** (0.75, 1, 2.5, 5, 10, or 25 μM) and a vehicle for different amounts of time (24, 48, and 72 h). Each sample was used in six parallel experiments. The samples were incubated in 5% CO_2_ at 37 °C for 48 h, and then 15 μL of MTT solution (5 mg/mL, 0.5%MTT) was added to each well in 150 μL of medium for 4 h at 37 °C. After that, 100 μL of 10% SDS solution (pH = 2) was added to each well for a terminal reaction for MTT. The absorbance was measured with a multiscan spectrum device (SpectraMax Paradigm, San Jose, USA) at wavelengths of OD 570 and 650 nm.

The CCK8 assay determined cell viability. The cell number in the logarithmic phase was 3 × 10^5^–6 × 10^5^/mL. It was treated with **8a** (0, 1, 2, 3, 5, and 10 μM) and a vehicle for different amounts of time (12, 16, 20, and 24 h). Each sample was used in six parallel experiments. The samples were incubated in 5% CO_2_ at 37 °C, and then 10 μL of CCK8 solution was added to each well in 100 μL of medium for 4 h at 37 °C. The absorbance was measured with a multiscan spectrum (SpectraMax Paradigm, San Jose, USA) at a wavelength of OD 450 nm.

### 4.3. Cell Co-Localization

The appropriate number of cells were taken and evenly placed into the 12-well plate, which had circular glass in the bottoms of the wells, in 5% CO_2_ at 37 °C. Then, tissue fixative solution was added, along with DAPI (Beyotime; C1005), DIL (Beyotime; C1036) staining solution. After washing 3 times, an anti-fluorescence quenching agent was added to the slides, covering the slides. We fixed the glass slides with nail polish (avoiding light for the whole process). We used a fluorescent microscope (Axio Observer 5 with ApoTome, Jena, Germany) for measurements.

### 4.4. Western Blot

The cells were cultured in a six-well plate, with the number of cells accounting for about 30–40% of the well plate. The concentration of 8a was set (0.5, 1, 2, 4 μM), 0.1% DMSO as the control group, and each concentration is set to three parallel, and incubated in a 37 °C 5% CO_2_ cell incubator for 48 h. Whole-cell proteins were extracted in 300 μL aliquots (total protein about 300 μg). RIPA lysis buffer with 5% protease was added along with a 50X phosphatase inhibitor cocktail on ice for 30 min; the solution was centrifuged at 12,000 r/min for 10 min. We used sodium dodecyl sulfate-polyacrylamide gel electrophoresis to process equivalent amounts of protein and then transferred the protein to PVDF membranes at room temperature for 1 h. We diluted each antibody with a sealing solution at a ratio of 1:1000, and incubated the solution overnight with rotation at 4 °C. The secondary antibody was rotated and incubated at constant temperature for 2 h. We used TBST washing 3 times (10 min/time). Bands were visualized using an enhanced chemiluminescence system (Bio-Rad, Shanghai, China) [49]. DNMT1 (1:1000, 24206-1-AP), cytochrome C (1:1000, 66264-1-Ig), and VDAC (1:1000, 10866-1-AP) antibodies were from Proteintech (Wuhan, China). β-Actin (1:1000, AF5001), Bax (1:1000, AF1270), Bcl-2 (1:1000, AF0060), and Ac-H3 (Lys56) (1:1000, AF1684) antibodies were from Beyotime (Shanghai, China). Finally, respective blots were incubated with Super ECL Detection Reagent. Both blots were simultaneously acquired using ChemiDoc™ Touch Imaging System (Bio-Rad Laboratories, CA, USA). The grayscale analysis was completed using Image Lab, and then the data were counted by GraphPad Prism 8 (San Diego, CA, USA).

### 4.5. Apoptosis Assay

Annexin V-FITC Apoptosis Detection Kit (C1062; Beyotime). After U937 cells were treated with different concentrations of **8a** and 0.1% DMSO, they were centrifuged at 1000× *g* for 5 min; then we discarded the supernatant. Annexin V-FITC binding solution and PI staining solution were added to resuspend the cells, which were then incubated at room temperature (20–25 °C) in darkness for 20 min. PI single staining and Annexin V-FITC single staining were set up to adjust the baseline. Flow cytometry (CytoFlex S, Shanghai, China) was used to measure.

### 4.6. ATP Assay Detection

ATP Assay Kit (S0026; Beyotime). The cells were treated with different concentrations of **8a** (0, 1, 2, 3, 5, and 10 μM) for 30 h, and a certain proportion of cell lysate was added to the cells to make them fully lysed. Then, the cells were centrifuged at 12,000× *g* at 4 °C for 5 min, and the supernatant was extracted. We added 100 μL of ATP test solution to the 96-well plate and left it at room temperature for 3–5 min. Then, we added 20 μL of sample or standard to the well, mixed quickly, waited at least 2 s, and measured by Luminometer.

### 4.7. Mitochondrial Potentiometric Membrane Detection

Mitochondrial Membrane Potential Assay Kit with JC-1 (C2006; Beyotime). The number of cells was controlled at 1 × 10^5^–6 × 10^5^/mL. **8a** concentrations (0.5, 1, 2, and 4 μM), 1 μM CCCP, and 0.1%DMSO were added into the cell suspension and cultured in an incubator at 37 °C and 5% CO_2_ for 48 h; then, 0.5 mL JC-1 staining working solution was added. Additionally, after incubation at 37 °C for 20 min, the solutions were washed twice with JC-1 staining buffer (1X) and resuspended for analysis by fluorescence microscope. CCCP was added as a positive control and 0.1% DMSO as a negative control. We used a fluorescent microscope (Axio Observer 5 with ApoTome, Jena, Germany) to measure.

### 4.8. Reactive Oxygen Detection

Cells were plated in 96 well plates and were treated with **8a** (0.5, 1, 2, or 4 μΜ) and 0.1% DMSO for 48 h. The cells were then stained with 5 µM CellROX™ Deep Red Reagent by adding the probe to the complete media. The cells were then washed with PBS and analyzed on a SpectraMax fluorescence microplate reader at a wavelength of OD 665 nm.

### 4.9. Proteomics

The cell cultivation was completed in a 6 mm plate; the number of cells accounted for roughly 30–40% of an orifice. We used 2 μΜ of drug and 0.1% DMSO. Each concentration had triplicate experiments over 48 h. Cells were resuspended in lysis buffer combined with a 1 X PMSF proteinase inhibitor cocktail (Thermo Scientific, USA). The total protein concentration was measured using Thermo Scientific NanoDrop One (protein A280). Then, equal amounts of proteins (approximately 1 mg) were diluted with 450 µL of NH_4_HCO_3_ (50 mM) and added to Ultra filtration tubes (Amicon Ultra-4, PLGC Ultracel-PL ultrafiltration membrane, 10 kDa), and we added a 10 mM DTT reduction (A2948; AppliChem), 20 mM IAA alkylation (I2273; Sigma). Each sample was centrifuged at 14,000× *g* to pass the solution through a filter. The resuspension was added before the total intracellular protein was hydrolyzed by trypsin (90057; Thermo Scientific™) at 37 °C (12–14 h) and desalted using a Thermo Scientific Pierce C18 tip. We freeze-dried samples and used 1% TFA resuspensions (see Appendix A for details). They were analyzed by mass spectrometry (Orbitrap Fusion Lumos, Shanghai, China). Screening for significantly different proteins was analyzed by DAVID (https://david.ncifcrf.gov (accessed on 12 March 2021)), STRNG (https://string-db.org (accessed on 12 March 2021)), and Cystoscope [50].

### 4.10. Molecular Docking

Molecular docking study of small molecules with DNMT1 X-ray structure (PDB ID: 4WXX) was conducted using Discovery Studio 2020 (Omaha, Nebraska) CDOCKER method based on the CHARMM all-atom force field. The range of 4 Å around the ligand of protein complex was used as the docking active pocket. Other parameters in the docking method adopted the default parameters of the software.

### 4.11. Quantitative RT-PCR

U937 cells were treated with different concentrations of **8a** (0.5, 1, 2, and 4 μM) with 0.1% DMSO as a control for 48 h. We used the simple RNA Total RNA Kit (Tiangen, Beijing, China) to extract from the sample. Additionally, we synthesized cDNA using a FastQuant RT kit (Tiangen, Beijing, China). Quantification of mRNA was carried out on a CFX Connect Real-Time PCR Detection System (Bio-Rad, Shanghai, China) using the SuperReal PreMix Plus (SYBR Green) kit (Tiangen, Beijing, China). The relative expression level of each candidate gene was calculated using GAPDH as the internal normalized control with the same calibrator.

DNMT1 forward primer: 5′-AGGCGGCTCAAAGATTTGGAA-3′

DNMT1 reverse primer: 5′-GCAGAAATTCGTGCAAGAGATTC-3′

GAPDH forward primer: 5′-CTGGGCTACACTGAGCACC-3′

GAPDH reverse primer: 5′-AAGTGGTCGTTGAGGGCAATG-3′

### 4.12. Binding Specificity and Affinity Assay

The specificity assay was performed by biolayer interferometry (BLI) using the Octet K2 System (ForteBio, Ulm, Germany). DNMT1 protein (188.2 KD) was purchased from the Active Motif Company. DNMT1 was diluted to 50 μg/mL using running buffer (PBS, 0.02% Tween-20) and loaded onto NTA biosensors for 600 s. **8a** was diluted in a concentration gradient (400, 200, 100, 50, and 25 μM) using a running buffer. After loading, the biosensors were baselined for 60 s, mixed with **8a** for 150 s, and then dissociated in the buffer for 100 s. The baseline-corrected binding curves were analyzed, and the equilibrium dissociation constants were obtained using software provided by ForteBio (Data Analysis 11.0).

### 4.13. Statistical Analysis

Data were reported as the means ± S.D. One-way ANOVA was performed with the statistical packages of GraphPad Prism 8 and OriginPro 8 (OriginLab Corporation, Northampton, USA). Additionally, we considered the difference of *p* < 0.05 significant.

## 5. Conclusions

In conclusion, **8a**, as an acridine lead compound, can target HDAC1 and DNMT1 to induce cell apoptosis. In vitro experiments showed that **8a** could effectively inhibit HDAC1 activity and induce cell apoptosis through various pathways. At the same time, **8a** was closely related to epigenetic regulation related genes, and we found that **8a** could interact with DNMT1 and affect the expression of DNMT1. Therefore, we hypothesized that **8a** inhibits epigenetic silencing of tumor genes (TSGs) through DNA methylation and histone alterations. However, the specific mechanism of action needs further study. This project’s development will be conducive to discovering highly effective non-nucleoside inhibitors and providing a theoretical basis for further research of antineoplastic drugs targeting HDAC1/DNMT1.

## Figures and Tables

**Figure 1 ijms-22-05516-f001:**
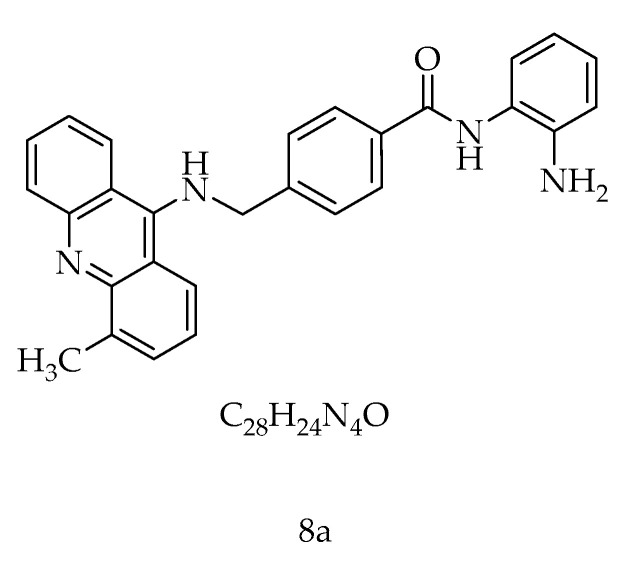
The structure of **8a** (*N*-(2-aminophenyl)-4-(((4-methylacridin-9-yl) amino) methyl) benzamide).

**Figure 2 ijms-22-05516-f002:**
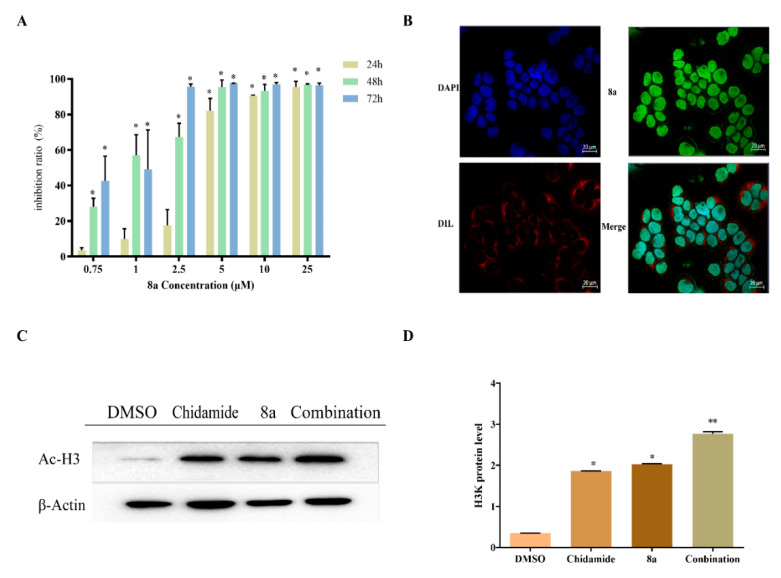
(**A**) Cell viability through use of an MTT assay. U937 cells were treated with different concentrations of **8a** (0.75, 1, 2.5, 5, 10, and 25 μM) for different amounts of time (24, 48, and 72 h). (**B**) Cell co-localization. U937 cells were cultured for 48 h with 2 μM of **8a**, and its emission peak is 476 nm. The cell nuclei were stained with 4′,6-diamidino-2-phenylindole (DAPI), and the membranes were stained with 1,1′-dioctadecyl-3,3,3′,3′-tetramethylindocarbocyanine perchlorate (DIL). Samples were observed under a 20x aperture through a fluorescent microscope (Axio Observer 5 with ApoTome). (**C**,**D**) Ac-H3 Western blots. U937 cells were treated with **8a** (2 μM), chidamide (0.5 μM), or their combination for 48 h, and Western blots analysis was used to evaluate the levels of Ac-H3; 0.1% dimethyl sulfoxide (DMSO) was used as a negative control. (* *p* < 0.05, ** *p* < 0.01).

**Figure 3 ijms-22-05516-f003:**
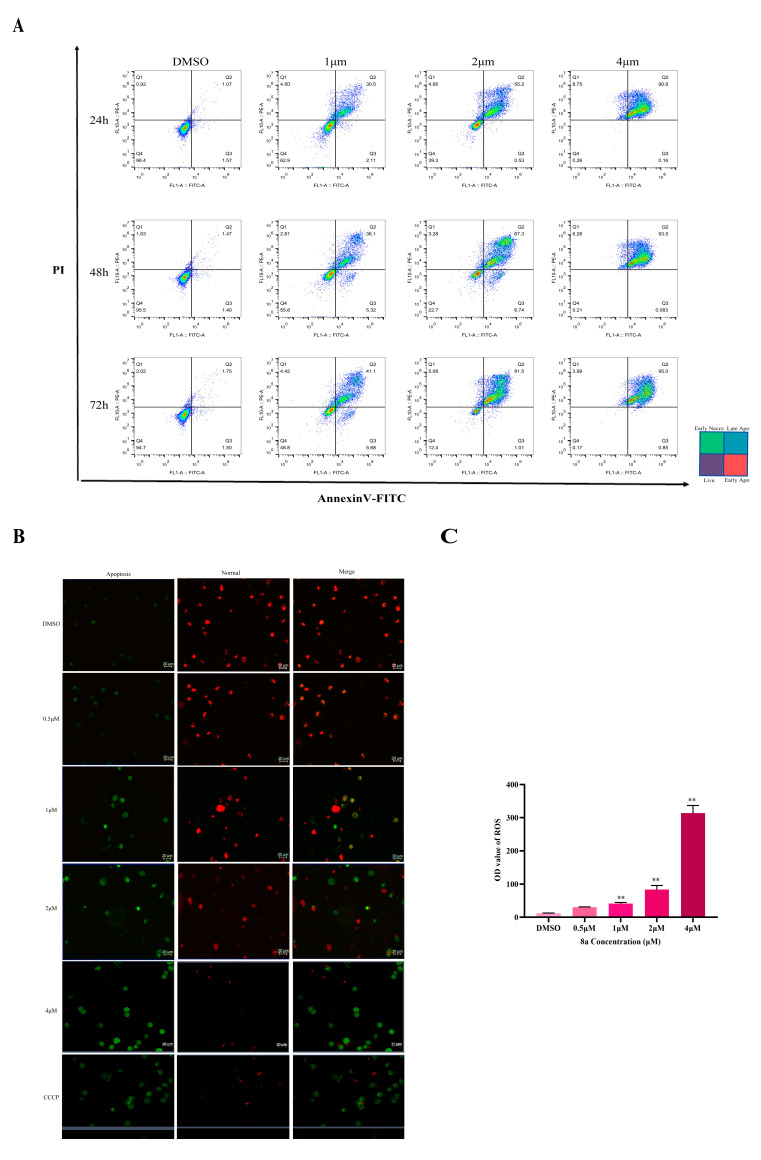
(**A**) Annexin V-FITC apoptosis assay. U937 cells were treated with different concentrations of **8a** (1, 2, and 4 μM) for different lengths of time (24, 48, and 72 h). Propidium iodide can stain necrotic cells or cells that have lost their cell membrane integrity in the late stage of apoptosis red fluorescence. Annexin V-FITC can enter the cytoplasm and bind to phosphatidylserine located on the inner side of the cell membrane, leading to the necrotic cells presenting green fluorescence; 0.1%DMSO was used as a control. (**B**) Mitochondrial membrane potential. Normal cells’ red fluorescence was reduced, and the green fluorescence of apoptotic cells was enhanced as the concentration of **8a** increased (0.5, 1, 2, and 4 μM); Carbonyl cyanate-3-chlorophenylhydrazone (CCCP) was the positive control, and 0.1%DMSO was used as the negative control. The normal cells showed red fluorescence, and the apoptotic cells showed green fluorescence. (**C**) Reactive oxygen detection. CellROX Deep Red Reagent probe was used to detect the changes of ROS in tumor cells induced by **8a**. U937 cells were treated with different concentrations of **8a** (0.5, 1, 2, and 4 μM) for 48 h (** *p* < 0.01); 0.1%DMSO was used as a control.

**Figure 4 ijms-22-05516-f004:**
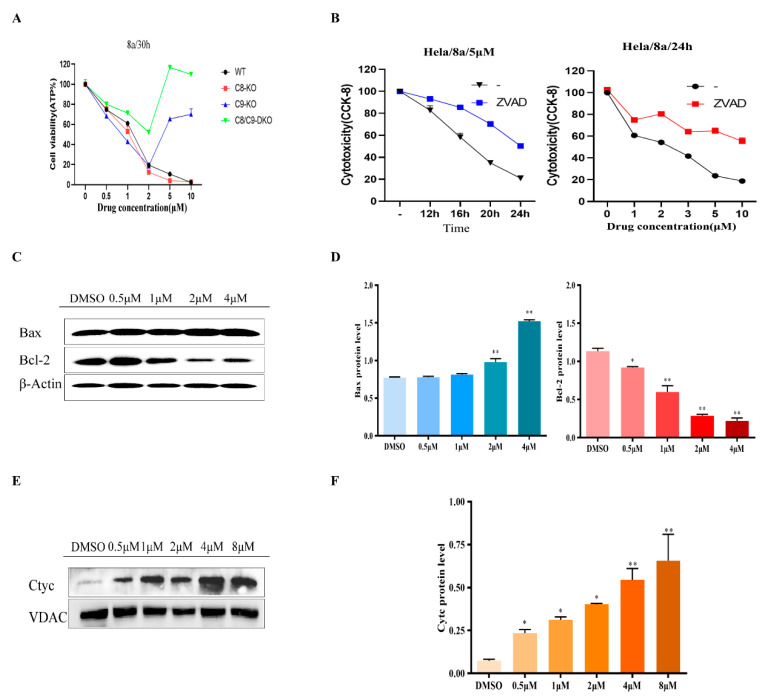
(**A**) **8a** affects the activity of gene knockout Hela cells. *caspase*-8 (C8) and *caspase*-9 (C9) knockdown cell lines were treated with 5 μM of **8a** for 30 h; cell viability was found by measuring adenosine 5′-triphosphate (ATP). (**B**) U937 cells were treated with Z-VAD-FMK and then treated with different concentrations of **8a** (0, 1, 2, 3, 5, and 10 μM) for 24 h. Another experiment used 5 μM of **8a** for different periods of time (12, 16, 20, and 24 h), respectively. Cell activity was determined by CCK8 assay. (**C**,**D**) Basal expression of apoptotic protein markers in **8a**-treated U937 cell lines. U937 cells were treated with different concentrations of **8a** (0.5, 1,2, 5, and 10 μM) for 48 h. Protein was extracted and Western blotted with antibodies against Bax, Bcl-2 and β-actin. 0.1% DMSO was used as a negative control. Western blots were quantified, and data are presented as means ± SD of three independent experiments (*n* = 3) (* *p* < 0.05, ** *p* < 0.01). (**E**,**F**) U937 cells were treated with different concentrations of 8a (0.5, 1,2, 5, and 10 μM) for 48 h. Protein was extracted and Western blotted with antibodies against Cyt C, and VDAC. 0.1% DMSO was used as a negative control. Western blots were quantified, and data are presented as means ± SD of three independent experiments (*n* = 3) (* *p* < 0.05, ** *p* < 0.01).

**Figure 5 ijms-22-05516-f005:**
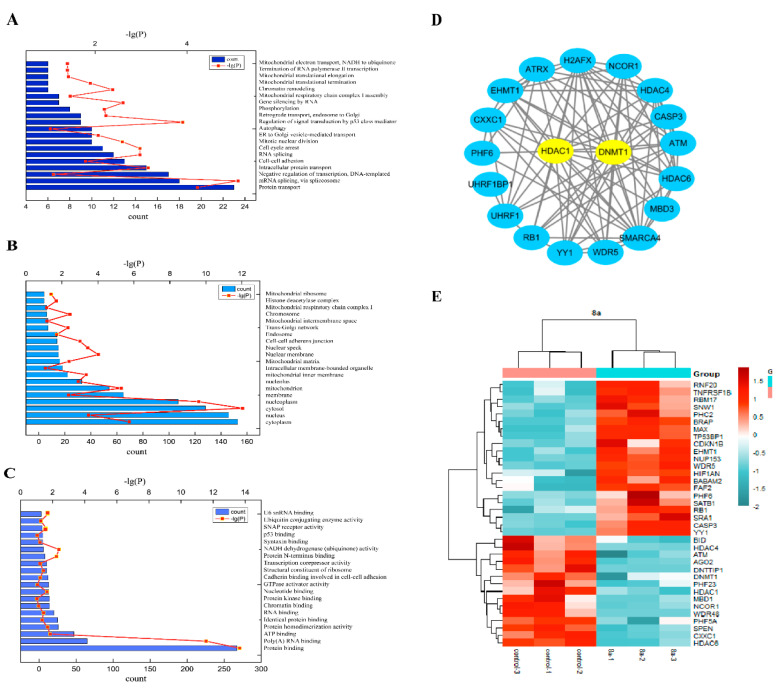
Proteomics analysis. (**A**–**C**) The BP (biological process), CC (cell component), and MF (molecular function) of GO were analyzed. (**D**) Protein–protein interactions to analyze proteins using STRING and cystoscope software. A total of 19 genes were associated with both histone deacetylase 1 (HDAC1) and DNA methyltransferase 1 (DNMT1). (**E**) A heatmap to analyze differentially expressed proteins down-regulated genes in blue and up-regulated genes in red.

**Figure 6 ijms-22-05516-f006:**
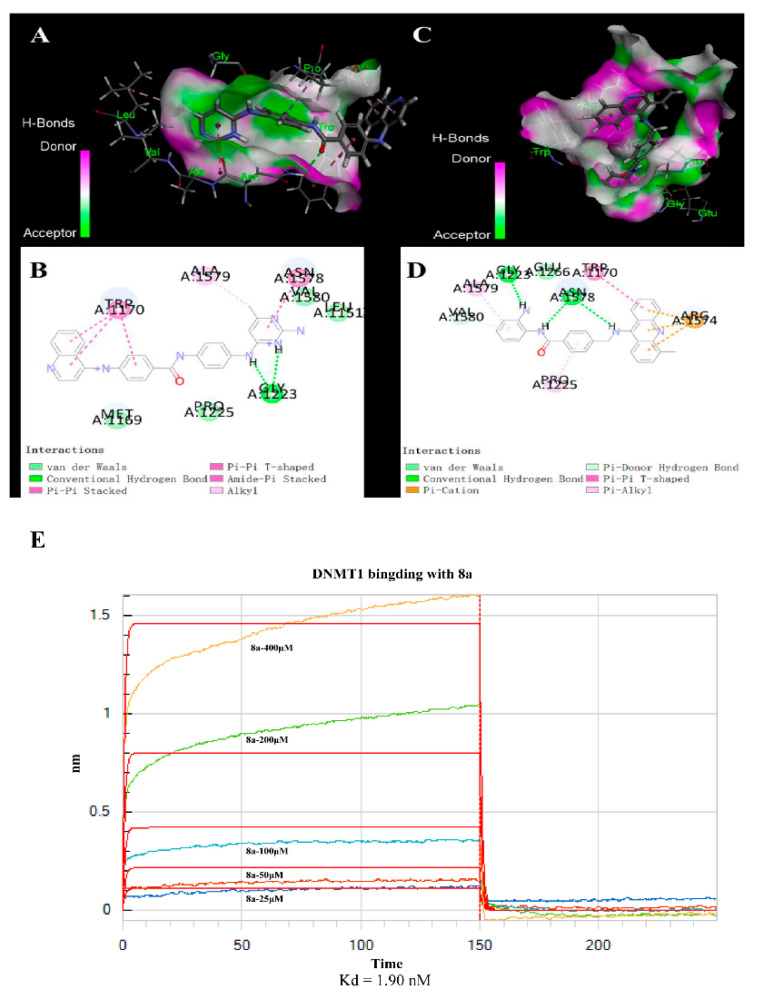
Molecular docking and interactions. (**A**–**D**) Docking modes of SGI-1027 (**A**,**B**) and **8a** (**C**,**D**) with DNMT1. Semi-flexible docking: 3D and 2D docking diagrams made using Discovery Studio 2019 software. (**E**) The binding affinity between **8a** and DNMT1 is based on biolayer interferometry (BLI). For DNMT1 combined with different concentrations of **8a** (25, 50, 100, 200, and 400 μM), the binding kinetics and dynamics were measured using an Octet K2 System (Forte Bio). The Kd (dissociation constant) value between **8a** and DNMT1 protein was 1.90 nM.

**Figure 7 ijms-22-05516-f007:**
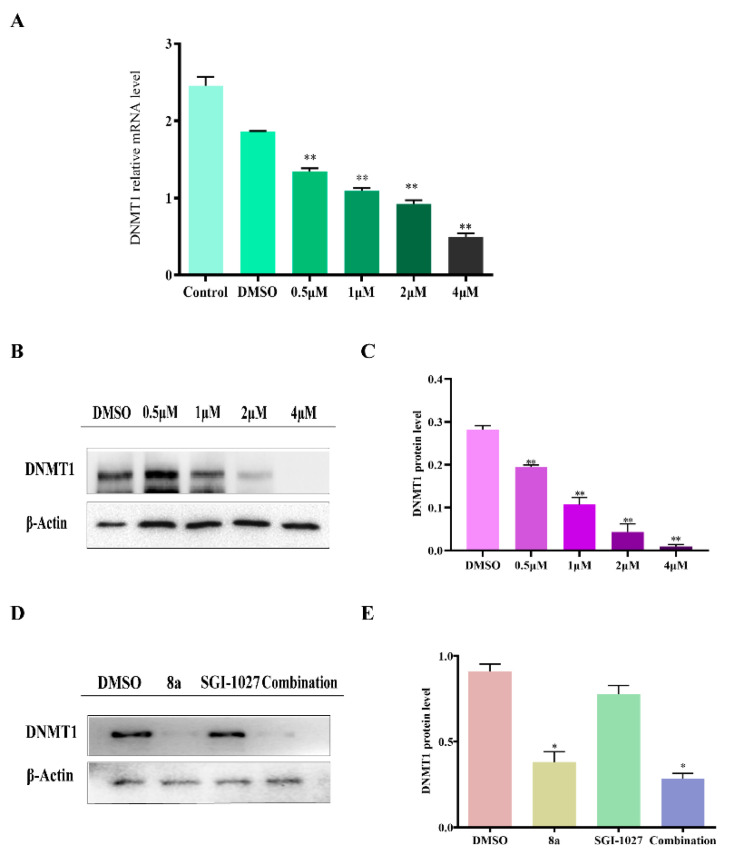
**8a** affected the expression of DNMT1. (**A**) RT-PCR was used to analyze the mRNA expression of *DNMT1* under different concentrations of **8a** (0.5, 1, 2, and 4 μM) in U937 cells (** *p* < 0.01); 0.1% DMSO can be used to eliminate the interference of compound solvents. (**B**,**C**) Western blots analysis of whole-cell extracts from cells treated with **8a** (0.5, 1, 2, and 4 μM) for 48 h (** *p* < 0.01); 0.1% DMSO was used as the control. (**D**,**E**) Western blots analysis of U937 cells treated with **8a** (2 μM) and SGI-1027 (3 μM) for 48 h. Whole-cell extracts from cells exposed to 0.1% DMSO was used as the control (* *p* < 0.05, ** *p* < 0.01).

## Data Availability

All data generated or analyzed during this study are included in this published article and its Appendix A.

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
