# Peer review of "8a, a New Acridine Antiproliferative and Pro-Apoptotic Agent Targeting HDAC1/DNMT1"

_ijms, 2021, doi:10.3390/ijms22115516_

Round 1
Reviewer 1 Report
The resubmitted manuscript, titled “8a, a new acridine antiproliferative and pro-apoptotic agent targeting HDAC1/DNMT1”, was ameliorated in the form and in concepts and many experiments were added, including other cell lines as suggested. However, some leaks were remained such as the validation of proteomic analysis by immunoblotting, which is discussed in terms of theory but only DNMT1 gene was validated, and the efficacy of 8a molecule as DNMT inhibitor respect to other molecule such as SGI-1027. Indeed, I didn’t see such difference in inhibitory effect between 8a and SGI-1027 (Fig. 7 D-E) like the authors affirmed. Moreover, I suggested the densitometric analysis of Western Blots, due to the small differences in protein expression levels and in the variations of housekeepings’ bands in the showed immunoblots. Nevertheless, none was performed.
In my opinion the authors should have tested the 8a antiproliferative activity also in the CCRF-CEM cells and K562.
I suggest to authors some minor revisions such as the increase of the quality/resolution of pictures in Figure 3, in particular figures 3A and 3B, because the text is unreadable and the specification of with which lines the proteomic experiment was performed, because this information is lacked.
Reviewer 2 Report
I read the revised version of the manuscript of Zhang and collaborators as well as the response to the three independent referees, which were complete and honest.
I believe that the authors did a fair revision work.
Author Response
We thank the reviewer for this precious comment. Thank you very much.
Reviewer 3 Report
The authors have tried to improve the paper taking into consideration the queries raised by the Reviewers. But many questions remain partially answered. As one example, this Reviewer has mentioned that the authors need to provide the Western blot protocol in detail. But in the revised manuscript, there is no mention of how much protein was loaded, about blocking or how the images were analyzed, etc. Also, I agree with Reviewer 2 that total H3 levels should be the proper control for H3K56 acetylation. I understand that sometimes antibodies do not work or are not available. But at least the authors could normalize it with Ponceau S stain?? This is just one example, and I can see many such errors throughout the manuscript. The Result section is too long with all the basic details making it difficult to read the manuscript.
Round 2
Reviewer 3 Report
No comments
This manuscript is a resubmission of an earlier submission. The following is a list of the peer review reports and author responses from that submission.
Round 1
Reviewer 1 Report
Manuscript ID: ijms-1138553
Title: 8a, a new acridine anticancer lead compound targeting HDAC1/DNMT1
Comments to the Author(s):
The manuscript by Zhang et al. describes the results of the experiments conducted using a histone deacetylase (HDAC) inhibitor, 8a. The authors mainly studied the efficacy of the compound, 8a, to induce apoptosis, by using different experimental approaches. Also, they studied the effect of the compound on DNMT1 by molecular docking and also by measuring the mRNA expression and protein levels. I have several major concerns, especially in the Methodology/ Results part of the manuscript. I am mentioning them below:
- In the Introduction, the authors based their hypothesis based on the fact that 8a is HDAC1 inhibitor but other studies (PMID: 32103894) have shown that 8a is a more potent inhibitor of HDAC2. This should be mentioned
- In the Methodology, the authors mentioned that the cell viability was measured using MTT assay and CCK8 cell counting kit. But in the Results (Figure 1A), showed the MTT assay alone. Why is this so? Also, they mentioned using nine different concentrations of 8a compound (0.1 to 25 µM) in the Methodology, but in the Results, Figure A shows the concentration starting from 0.75 µM to 25 µM, with no mention about the lower concentrations tested. The authors need to explain this.
- For the nuclear co-localization experiment (Methods sub-section 4.3 & Figure 1B), it is not clear what staining is used to obtain green fluorescence for the compound, 8a. There is no explanation for why 2 µM concentration of 8a was used for this experiment.
- Methodology sub-section 4.4, the protocol for Western blot is not explained correctly. There are no reference IDs for the antibodies used or their dilutions. The authors should provide the Western Blot methodology in detail. Also, they mentioned in the Methods and Figure legend (Fig. 1 C) that U937 cells were treated with different concentrations of 8a for 48h. But Figure 1C represents only one concentration of 8a with no mention of the concentration used. What was the concentration of Chidamide used? Please mention H3KAc instead of H3K and also the protein size. Again, the Results 2.1, the authors mentioned “It was showed that high expression of H3K………compare to a normal group” What do the authors mean by the normal group?
- For the Apoptosis-necrosis assay (Methods 4.5), were Annexin V-FITC/ PI or Hoechst/ PI used? What is CCCP (Methods 4.6)? Where are the Results of positive control? What is the staining used for red and green fluorescence in Fig. 2C? Because the protocol was not explained sufficiently, it is difficult to understand how the authors performed this assay. If a commercial kit was used, at least the reference ID should be provided to understand how the assay was performed.
- In the previous article published by the authors (Ref. No. 17), the same apoptosis/ necrosis assay was performed on U937 cells for a 2µM dose for 48 hours. But in this article, the results do not look similar for the same dose and timing?
- Results 2. 2, it is not clear how the authors analyzed the exogenous apoptotic pathway?
- Methodology sub-section 4.10, the authors mentioned that DNMT1 gene expression was normalized using beta-actin but provided GAPDH reverse and forward primer sequences?
- No mention of what is SGI-1027.
Reviewer 2 Report
Zhang and collaborator previously published a family of acridine-derived compounds with HDAC inhibitory activity.
Now, one of these compounds, named 8a, is studied further as it displays antiproliferative activity in lymphoma derived U937 cells.
Data are presented that demonstrate a double targeting of HDAC1 and DNMT1. 8a appears at an inhibitor of HDAC catalytic activity, while it putatively inhibits DNMT1 protein expression level.
This is well-performed investigation, but several important issues should be addressed.
- As a general commentary, I consider it premature to define 8a as an “anticancer” compound, as no in vivo study is performed. The authors should define the compound as an anti proliferative and pro-apoptotic agent.
- In figure 1C, the notion of anti H3 (acetyl?) blot should be reported more carefully. What are we seeing in this blot? Are the authors possibly referring to “Acetylated H3 on Lysine 9” or other? The antibodies used in the study are not mentioned and the western blotting procedures are not mentioned in the materials and methods section. Also, a proper control for acetylated histone H3 would be a blot against total H3
- Figure 6B should be redesigned as the bars of the points “DMSO” and 0.5 micromolar” are not visible
- Although already published, the chemical formula of 8a and the full IUPAC name of the compound should be presented here.
- It is unfortunate that, to highlight the effects of 8a, parallel administration of other (and less potent) molecules of the same acridine compounds series has not been performed.
- The manuscript needs a re-reading and extensive by a native English speaker. Throughout the manuscript the use of present/past tenses is often not appropriate generating quite some confusion in the reader. While the data reported are of interest, the language presentation is substandard.
- The authors state that supplementary materials are available, but I could not find these files in the submission.
Reviewer 3 Report
The results, presented by Zhang and colleagues in the manuscript “8a, a new acridine anticancer lead compound targeting HDAC1/DNMT1”, are interesting because they demonstrated that just one molecule can influence both DNA methylation and histone acetylation simultaneously. The article is well written and easy to understand, but in my opinion some improvements should be necessary. 1. I suggest to test the 8a molecule using another cell line at least (also another tumor type) in order to evaluate if the molecule can have the same pharmacological effect, independently by mutational state of U937 cells or histological subtype. Otherwise, the assertation of antineoplastic drug able to inhibit tumor’s growth is actually unsuitable. 2. Why didn’t the authors think to test the 8a molecule also on a leukemia cell line? In the introduction and discussion sections, they speculate about HDAC and DNMT1 expressions in this hematological cancer and not about the lung cancer. In my opinion, the text should disclose about the known background on the lung cancer’s epigenetic profile, seeing as it is the cell line used for the research. 3. The authors affirmed that proteome data analysis showed how 8a is able to act by “turning back on” silenced genes, among them some tumor suppressor genes, because of DNMT1 mRNA down-regulation. However, I didn’t find any RT-PCR validation on downstream genes of DNMT1 before and after cell treatment with 8a. In addition, none mention was made about the downstream genes of Ac-H3K up-regulation. 4. I found some incongruities between the H3K protein levels reported in the histogram and the immunoblot image. I suggest accomplishing the densitometric analysis, in particular for the H3K western blot, but recommended also for the others. 5. The WB data showed that 8a has a comparable effect with Chidamide or SGI-1027 and its combination with these drugs can enhance the effect of 8a as single agent. So, what the authors can conclude about these results? Can 8a be used as antineoplastic single agent? Are there published data about the advantage of use of acridine compounds in terms of toxicity and specificity, as alternative treatments of epigenetic agents actually in use or in clinical trials? In the light of the above-mentioned argumentations, I can’t suggest the publication of this paper in this form, except after integrating the data with supplementary analysis.